# Effects of Supplementation in Vitamin D3 Deficient or Insufficient Children with Allergic Diseases

**DOI:** 10.3390/medicina57101052

**Published:** 2021-10-01

**Authors:** Andjelka Stojkovic, Katerina Dajic, Jasmina Milovanovic, Slobodan M. Jankovic, Nenad V. Markovic, Andrijana Kostic

**Affiliations:** 1Pediatric Clinic, University Clinical Center Kragujevac, 34000 Kragujevac, Serbia; katerinadajic@gmail.com (K.D.); andrijanak88@yahoo.com (A.K.); 2Department of Pediatrics, Faculty of Medical Sciences, University of Kragujevac, 34000 Kragujevac, Serbia; 3Department of Pharmacology, Faculty of Medical Sciences, University of Kragujevac, 34000 Kragujevac, Serbia; jasminamilo@yahoo.com (J.M.); slobnera@gmail.com (S.M.J.); 4Department of Surgery and Toxicology, Faculty of Medical Sciences, University of Kragujevac, 34000 Kragujevac, Serbia; operacioni.centar@kc-kg.rs; 5General Surgery Clinic, University Clinical Center Kragujevac, 34000 Kragujevac, Serbia

**Keywords:** allergy, child, deficiency, dose, insufficiency, vitamin D

## Abstract

*Background and Objectives*: Although vitamin D insufficiency or deficiency is prevalent in children with allergic diseases, recommendations for supplementation dosing regimens are imprecise and variable in the literature, because clinical trials aiming to determine optimal doses were scarce in the past. This study aimed to investigate supplementation of vitamin D3 that may achieve therapeutically effective but not toxic serum levels in a subpopulation of children with allergic diseases and concomitant hypovitaminosis D. *Materials and Methods*: The retrospective, observational study with a cross-sectional design included 94 children suffering from allergic diseases and having vitamin D deficiency/insufficiency who were prescribed high-dose vitamin D3 supplementation by a pediatrician for at least 6 weeks and not more than 9 weeks. Serum levels of the major metabolite of vitamin D (25-(OH)D) were determined in all children twice: before and two weeks after the end of vitamin D3 supplementation. *Results*: An increase in serum level of the 25-(OH)D after supplementation was significant. However, if the subjects had higher serum levels of the 25-(OH)D before the supplementation, and if the supplementation lasted 8 instead of 6 weeks, the absolute increase in serum level of the 25-(OH)D was lower. Patients taking corticosteroids as inhalation or intranasally had a more intense effect of vitamin D3 supplementation, i.e., the absolute increase in levels of 25-(OH)D was higher than in patients not using such medication. *Conclusions*: Vitamin D deficiency and insufficiency in children with allergic diseases can be treated with maximal recommended doses of vitamin D3 for a short period of time, especially if they were prescribed with inhalation or intranasal corticosteroids.

## 1. Introduction

Vitamin D has many roles apart from the regulation of calcium homeostasis, one of which is the suppression of allergy [1]. The prevalence of hypersensitivity to at least one allergen in school-age children continues to grow and is approaching 50%. The International Study of Asthma and Allergy in Childhood (ISAAC) showed that the prevalence of asthma and allergic diseases in European countries ranges from 15 to 35%. Every fourth child in Serbia (25%) has at least one episode of wheezing in the first seven years of life, which puts a high burden on the health budget and family [2]. One of the risk factors for wheezing and allergic disease is hypovitaminosis D, which is prevalent among infants due to insufficient intake in diet [3].

There are numerous immunomodulatory effects of vitamin D, and some of them are directly related to an allergic reaction. Almost all cells of the immune system (T and B lymphocytes, neutrophils, macrophages, dendritic cells) express receptors for vitamin D and the enzyme 1-α hydroxylase. Vitamin D enhances cellular immune response (chemotaxis and phagocytic ability), and at the same time, reduces the excessive activation of the humoral immune response [4,5,6]. The safe serum level of the major metabolite of vitamin D (25-(OH)D) is below 100 ng/mL (250 nmol/L), with concomitant a calcium/creatinine clearance less than 0.4 [7]. Vitamin D achieves noncalcitropic (immunologic) effects if the serum level of the main metabolite of vitamin D is higher than 40 ng/mL (100 nmol/L), while for the calcitropic effect, much lower serum levels are sufficient (≤20 ng/mL or ≤50 nmol/L). Breast milk contains vitamin D in a concentration below 60 IU/L, which is insufficient to achieve minimum effective serum concentrations if an infant is only breastfed. The American Academy of Pediatrics recommends a daily intake of 400 IU of vitamin D3 for a breastfed infant or introduction of a milk formula enriched with vitamin D3 for at least some meals during a day [8]. However, about 40% of infants do not receive the recommended daily doses even in developed countries. This suggests that 40% of infants are probably in hypovitaminosis D, which increases the risk of developing rickets, allergic diseases, and other diseases associated with vitamin D deficiency [9,10]. Supplementation of vitamin D could be harmful if it has overdosed. Safe daily intake of vitamin D for prolonged periods, which may provide serum concentrations above 40 ng/mL (100 nmol/L) but below the toxic threshold of 100 ng/mL (250 nmol/L), is 10,000 IU/day. Serum levels of the major metabolite of vitamin D (25-(OH)D) above 150 ng/mL (375 nmol/L) were associated with acute hypercalcemia [7,11].

This study aimed to investigate supplementation of vitamin D3 that may achieve therapeutically effective but not toxic serum levels in a subpopulation of children with allergic diseases and concomitant hypovitaminosis D.

## 2. Patients and Methods

This retrospective, observational study with cross-sectional design included 94 children suffering from allergic diseases (asthma, allergic rhinitis, atopic dermatitis, urticaria, or food allergies), of both sexes, 1–17 years old, and who were prescribed high-dose vitamin D3 supplementation by a pediatrician for at least 6 weeks and not more than 9 weeks. The patients were treated from January to December 2017 in an outpatient clinic of University Clinical Center Kragujevac, Serbia. The study was approved by the Ethics Committee of the University Clinical Center Kragujevac (No 01/4965, 19 May 2016.) before its onset and registered in the Balkan Clinical Research Registry with the number BCRR-2021-003. Serum levels of the major metabolite of vitamin D (25-(OH)D) were determined in all children twice: before and exactly two weeks after the end of supplementation. Children were included if the following criteria were met: hypovitaminosis D, vitamin D3 supplementation prescribed by a pediatrician for 6–9 weeks during the year 2017, and children with one of the following diseases: asthma, allergic rhinitis, atopic dermatitis, urticaria, food allergy. The exclusion criteria were children with hypovitaminosis D who had other associated diseases of the liver, kidneys, gastrointestinal tract, hematopoietic and skeletal systems, children on a special diet, children on vitamin D supplementation less than 6 weeks before the moment when a pediatrician prescribed the last supplementation regimen of 6–9 weeks, those from whom reliable data could not be obtained, and children with a high risk of side effects of vitamin D3 therapy (see the Figure 1: Flow diagram).

Presumed independent variables followed in the study were (1) a season when serum 25-(OH)D3 level was measured (spring, autumn, or winter); (2) vitamin D supplementation that ended the latest 6 weeks before the onset of the last vitamin D3 supplementation regimen prescribed by a pediatrician; (3) whether foods rich or enriched with vitamin D were present in diet: fish, fish oil, egg yolks, mushrooms, milk, and dairy products, butter; (4) comorbidities (one of the following: allergy to cow’s milk, rickets, obesity). The following confounders were also taken into account: age of a child, gender, body weight, body height, body mass index, exposure to the sun, skin prick test on 20 inhalation and 20 nutritive allergens (positive/negative), and socioeconomic status.

The study participants were divided into groups based on serum levels of 25-(OH)D3 measured before the onset of vitamin D3 supplementation. Children with serum levels above 30 ng/mL were considered *vitamin D sufficient*; those with serum levels between 20 and 30 ng/mL were looked at as *vitamin D insufficient*. If pre-supplementation serum levels were between 10 and 20 ng/mL, the children were classified as *vitamin D deficient*, while the group with *severe vitamin D deficiency* was composed of children with serum levels below 10 ng/mL. All children except those vitamin D sufficient were prescribed oral vitamin D3 supplementation 4000 IU/day; however, the duration of supplementation was shorter for vitamin D insufficient children (6 weeks) than for vitamin D deficient children (8–9 weeks, regardless of severity).

Measurement of total serum 25-(OH)D levels were made by electrochemiluminescence binding assay (ECLIA) on Cobas^®^e 601 Automatic Analyzer (Roche Diagnostics, Mannheim, Germany) in Central laboratory of UCC Kragujevac.

### Statistics

Continuous data were described by mean and standard deviation, and data normality was checked by Shapiro–Wilk’s test. To test the hypothesis that there were significant differences between mean values of continuous variables with normal distribution, we used Student’s *t*-test for dependent samples or one-way analysis of variance (ANOVA). Otherwise, we used respective nonparametric tests (Wilcoxon’s matched pair rest or Kruskal–Wallis nonparametric analysis of variance). Only if the probability of the null hypothesis was equal to or below 0.05 were the differences considered statistically significant.

The sample size was calculated using G*Power 3.1.9. software, where input parameters were minimal statistical power of 99%, probability of type 1 error (α) 0.01 for two-side hypothesis testing, and effect size based on results of the study by Talib et al. in which mean serum level of 25-(OH)D measured before the onset of vitamin D3 supplementation was 13.0 ± 3.9 ng/mL, and after vitamin D supplementation with 4000 IJ, it increased to 34.0 ± 14.3 ng/mL [12]. Based on the above parameters, using a *t*-test, the required sample size of at least 15 patients was calculated. Statistical data processing was performed using standard statistical software IBM SPSS statistic version 18.0 (SPSS Inc., Chicago, IL, USA).

## 3. Results

This observational cohort study included initially 110 subjects; however, since 16 of them were either noncompliant or did not have a second measurement of 25-(OH)D, only 94 children from the initial cohort were included in the final dataset further statistically processed. There were 50 (53.2%) boys and 44 (46.8%) girls in the final dataset. The average age of the study participants was 8.3 ± 3.9 (range 1–17) years. Other characteristics of the study participants are shown in Table 1. Before the vitamin D3 substitution therapy, a total of 68 (72.3%) study subjects had vitamin D deficiency and severe vitamin D deficiency.

Serum levels of 25-(OH)D before the vitamin D3 supplementation (at baseline) were not significantly different among the age groups (F = 0.166, *p* = 0.847); the same was observed after the supplementation of vitamin D3 (F = 1.348, *p* = 0.265) (Table 1 and Table 2). Increase in serum levels of 25-(OH)D after supplementation was not significantly different (Student’s *t*-test = 0.841, *p* = 0.406) among the groups of children treated for 6 (13.66 ± 14.68 ng/mL) and 8 weeks (16.38 ± 12.13 ng/mL) with vitamin D3 (Table 3). Girls had a slightly better response (mean after/before difference = 18.25 ± 13.90 ng/mL) to vitamin D3 substitution therapy than boys (mean after/before difference = 13.31 ± 11.52 ng/mL), but this difference was not significant (Student’s *t*-test = −1.860, *p* = 0.066). If the whole study sample is taken into account, regardless of the duration of vitamin D3 supplementation, mean serum vitamin D level before vitamin D3 substitution was 16.83 ± 5.6 ng/mL, and after the supplementation, it rose to 32.46 ± 12.7 ng/mL (Student’s *t*-test = 14.228, *p* = 0.000).

After completion of vitamin D3 substitution therapy with the dose of 4000 IU/day, sufficient serum levels of 25-(OH)D (>30 ng/mL) were reached in 48 (51%) patients, while the remaining 46 (49%) subjects were below this cutoff point. The mean serum level of 25-(OH)D in the group that surpassed the cutoff point for vitamin D sufficiency was 44.06 ± 8.10 ng/mL, but only 16 (17%) subjects among them had 25-(OH)D serum levels over 40 ng/mL.

A multiple linear regression model was built with an increase in the 25-(OH)D serum levels after a period of supplementation as the outcome (dependent variable). Effects of independent variables on the outcome were adjusted for the other confounders entered in the model: age, sex, a season when serum 25-(OH)D3 level was measured, vitamin D supplementation that ended the latest 6 weeks before the onset of the last vitamin D3 supplementation regimen prescribed by a pediatrician, serum level of 25-(OH)D before supplementation, duration of supplementation in weeks, serum level of IgE, prescription of inhalation corticosteroids (yes/no), prescription of leukotriene antagonists (yes/no), prescription of intranasal corticosteroids (yes/no), skin prick test on 20 inhalation and 20 nutritive allergens (positive/negative), and calendar month of hospitalization. The model was significant (F = 2.833, *p* = 0.006) and explained almost a third of the outcome variability (R^2^ = 0.328, R^2^_adjusted_ = 0.212). The model satisfied assumptions of the multiple linear regression: normal distribution of residuals, homoscedasticity, lack of collinearity, and significant linear relationship between each of the continuous predictors and the outcome. The following predictors showed significant influence on increase in the 25-(OH)D serum levels after period of supplementation: serum level of the 25-(OH)D before supplementation (B = −0.902, 95% CI −1.666 to −0.138, *p* = 0.022), weeks of supplementation (B = −7.251, 95% CI −12.805 to −1.697, *p* = 0.011), prescription of inhalation corticosteroids (B = −8.132, 95% CI −15.096 to −1.167, *p* = 0.023), and prescription of nasal corticosteroids (B = −6.622, 95% CI −12.526 to −0.718, *p* = 0.029). Children prescribed with inhalation corticosteroids (*n* = 36) had higher increase in the the 25-(OH)D levels than children without (*n* = 58) such prescription (19.3 ± 15.4 ng/mL vs. 13.4 ± 10.5 ng/mL, respectively). Children prescribed with nasal corticosteroids (*n* = 37) had higher increase in the the 25-(OH)D levels than children without (*n* = 57) such prescription (17.4 ± 14.7 ng/mL vs. 14.5 ± 11.5 ng/mL, respectively).

## 4. Discussion

Although in our study, daily supplementation doses were at the upper limit of doses recommended by official summaries of product characteristics (SmPCs) [13], none of the study subjects reached postsupplementation serum concentrations of the 25-(OH)D above 80 ng/mL, which is far below minimum hypercalcemic serum concentration of the 25-(OH)D, i.e., 100 ng/mL [14]. Increase in serum level of the 25-(OH)D after supplementation was significant, regardless of whether the supplementation lasted 6 or 8 weeks. However, if the subjects had higher serum levels of the 25-(OH)D before the supplementation, the absolute increase in serum level of the 25-(OH)D was lower. Patients taking corticosteroids as inhalation or intranasally experienced a more intense effect of vitamin D3 supplementation, i.e., the absolute increase in levels of 25-(OH)D was higher than in patients not using such medication.

Prescribing the right dose for treatment of vitamin D deficiency or insufficiency remains a challenging task in the everyday clinical practice of a physician: no clear and sufficiently precise recommendations exist; a recent study found 350 different dosing regimens prescribed by physicians in France to patients of all ages. The situation is even more difficult for pediatricians [15]. In general, doctors avoid prescribing maximal recommended doses, fearing from potential toxicity of this liposoluble vitamin for which cases of hypervitaminosis were described in the past [16]. Our study showed that maximum recommended doses do not reach toxic levels of 25-(OH)D in serum while successfully correcting states of deficiency and insufficiency. There is rather a large therapeutic window for vitamin D3, and daily doses of 4000 IU could be considered safe and effective; when faced with a case of vitamin D deficiency or insufficiency, it is much more cost-effective to treat the patient with the maximum recommended doses—probability of attaining target serum levels of 25-(OH)D is higher, while the risk of toxicity is not increased. Patients with lower pretreatment serum levels of 25-(OH)D benefit more from vitamin D3 supplementation since an increase in serum levels is much higher after therapy, and respective improvement in bodily functions will be larger.

Our study showed that patients using inhalation or intranasal corticosteroids had higher postsupplementation serum levels of 25-(OH)D, which could be explained by the effects of corticosteroids on the transformation of vitamin D3 to 25-(OH)D and probably by increased bone turnover and relative hypercalcemia induced by corticosteroids, which affect the creation of an active form of vitamin D3 [17]. Although inhalation and intranasal corticosteroids are absorbed less than their orally administered counterparts, they still reach the blood in a significant amount and may have systemic effects [18]. It seems a plausible conclusion that children on therapy with inhalational or intranasal corticosteroids, when treated with maximum recommended doses of vitamin D3 (if having vitamin D3 deficiency or insufficiency), will have a reliable therapeutic response. A similar practice was already accepted by the majority of pediatric rheumatologists when prescribing systemic corticosteroids [19].

There are certain limitations of our study. In the first place, we used only one supplementation dose of vitamin D3, making the comparison of efficacy and safety with somewhat lower doses impossible (e.g., lower doses may have achieved similar serum levels of 25-(OH)D with the same safety, but this remains unknown). Second, our sample size was relatively small, analyzing subgroups impractical due to insufficient statistical power. Therefore, our results should be considered preliminary, and larger studies are necessary to confirm them in the future.

In conclusion, vitamin D deficiency and insufficiency in children with allergic diseases can be treated with maximal recommended doses of vitamin D3 for a short period of time, especially if they were prescribed with inhalation or intranasal corticosteroids.

## 5. What Is New?

Substitution dose of vitamin D3 must be greater than 4000 IU/daily and the duration of substitution longer than two months for children with allergic disease and concomitant hypovitaminosis D to achieve serum vitamin D level 40 ng/mL and more.

## Figures and Tables

**Figure 1 medicina-57-01052-f001:**
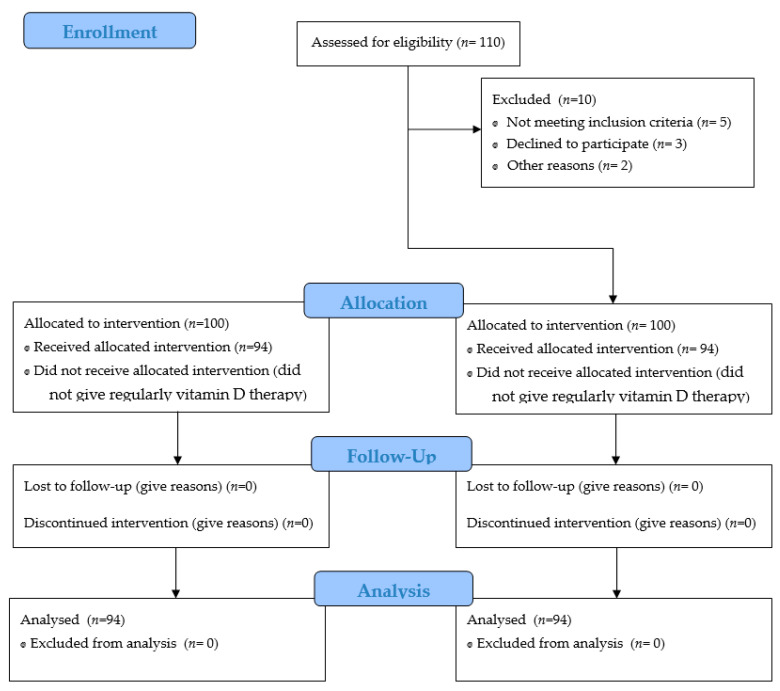
Flow diagram.

**Table 1 medicina-57-01052-t001:** Baseline serum levels of 25-(OH)D before vitamin D3 supplementation.

Variable		Number of Subjects, *n* (%)	Mean Serum 25-(OH)D Level (ng/mL) ±SD before Therapy
Age groups	Toddlers (1–3 years)	9 (9.6%)	17.0 ± 7.2
	Preschool children (4–6 years)	28 (29.8%)	16.3 ± 6.6
	School children (7–18 years)	57 (60.6%)	17.0 ± 4.8
Degree of the vitamin D deficit	Severe deficiency (<10 ng/mL)	11 (11.7%)	7.5 ± 2.2
	Deficiency (10–20 ng/mL)	57 (60.6%)	15.3 ± 2.7
	Insufficiency (20–30 ng/mL)	26 (27.7%)	23.9 ± 2.0
	Total	94 (100%)	16.8 ± 5.6
Allergic diseases that children had	Asthma	5 (5.3%)	17.8 ± 3.1
	Allergic rhinitis	4 (4.3%)	15.3 ± 2.8
	Atopic dermatitis	4 (4.3%)	12.9 ± 7.7
	Asthma + allergic rhinitis	65 (69.1%)	18.2 ± 5.4
	Asthma + allergic rhinitis + multiple food allergies	15 (16.0%)	12.0 ± 4.1
	Asthma + allergic rhinitis + atopic dermatitis	1 (1.1%)	14.20±/

**Table 2 medicina-57-01052-t002:** Serum levels of 25-(OH)D3 after vitamin D supplementation.

Variable		Number of Subjects, *n* (%)	Mean Serum 25-(OH)D Level (ng/mL) ±SD after Therapy
Age groups	Toddlers (1–3 years)	9 (9.6%)	34.1 ± 11.6
	Preschool children (4–6 years)	28 (29.8%)	35.4 ± 16.0
	School children (7–18 years)	57 (60.6%)	30.7 ± 10.8
Degree of the vitamin D deficit	Severe deficiency (<10 ng/mL)	1 (1.1%)	5.90±/
	Deficiency (10–20 ng/mL)	13 (13.8%)	17.7 ± 2.0
	Insufficiency (20–30 ng/mL)	32 (34%)	26.5 ± 3.0
	Sufficiency (>30 ng/mL)	48 (51.1%)	44.0 ± 8.1
	Total	94 (100%)	32.4 ± 12.7
Allergic diseases that children had	Asthma	5 (5.3%)	34.07± 12.1
	Allergic rhinitis	4 (4.3%)	25.0 ± 5.6
	Atopic dermatitis	4 (4.3%)	29.4 ± 6.2
	Asthma + allergic rhinitis	65 (69.1%)	33.1 ± 12.0
	Asthma + allergic rhinitis + multiple food allergies	15 (16.0%)	32.1 ± 14.9
	Asthma + allergic rhinitis + atopic dermatitis	1 (1.1%)	23.00±/

**Table 3 medicina-57-01052-t003:** Mean serum levels of 25-(OH)D before and after substitution therapy with 4000 IU/day of vitamin D3 during 6 and 8 weeks, respectively.

Duration of Substitution with Vitamin D3	Number of Subjects, *n* (%)	Mean Serum 25-(OH)D Levels (ng/mL)		*p*-Value
		Before Substitution	After Substitution	
6 weeks	26 (27.7%)	23.95 ± 1.9	37.61 ± 14.89	0.000
8 weeks	68 (72.3%)	14.10 ± 3.9	30.48 ± 11.34	0.015

## Data Availability

The data supporting reported results can be found at Faculty of Medicine, University of Kragujevac: send requests to: sjankovic@medf.kg.ac.rs.

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
