# Peer review of "Effects of Supplementation in Vitamin D3 Deficient or Insufficient Children with Allergic Diseases"

_medicina, 2021, doi:10.3390/medicina57101052_

Round 1
Reviewer 1 Report
This paper describes Vitamin D levels before and after oral supplementation in Vitamin D3 in allergic children.
Comments
The title is not appropriate. This paper does not defines the optimal therapeutic dose as only one dose and 2 timings are assessed. Also there are, in this paper, no physiological read out that defines if the achieve concentrations are optimal. The fact the achieved concentrations are below 80 ng/mL is not sufficient to define these concentrations as optimal. The optimum 25(OH)VTD serum concentration is still controversial.
It is misleading to compare timing of supplementation as different timings were applied to different populations. Conclusion on this point are doubtful.
If data on corticosteroid are a key conclusive point of the study, it would be better to describe them more in details. It could be interesting to have the number of children in these groups and the 25-(OH)D levels before and after supplementation.
Conclusions seem overstated. It would be better to write “children with allergic diseases can be treated with maximal recommended doses of vitamin D3 for a short period of time, especially…”
It would be easier to read if table 1 and 2 would be presented as a unique table with 25-(OH)D levels before and after the supplementation on the same line.
Elecsys kits on cobas do not measure 25(OH)VTD3 but the total concentration of vitamin D (D2 + D3). Therefore, terminology should be adjusted and the abbreviation 25-(OH)D should be favored.
The level of toxicity is sometimes defined as 100 ng/mL (line 64) and sometimes as 110 mcg/mL. For ease of reading, it is better to keep the same cut-off for the entire paper.
The dose of Vitmin D is referred as 5,000IJ line 124 whereas it is referred as 4,000IU/day for the entire paper.
Line 227 did the author wanted to write “sample” or sample size”?
Author Response
University of Kragujevac,
Faculty of Medical Sciences
To: Editors of Medicina
Dear Editors,
we are submitting our manuscript No medicina-1389149, entitled “OPTIMAL THERAPEUTIC DOSE IN VITAMIN D3 DEFICIENT OR INSUFFICIENT CHILDREN WITH ALLERGIC DISEASES” corrected according to the requests of the reviewers. The corrections in the manuscript are marked with “track changes” option. Below are corrections listed point-by-point according to requests of the reviewers, marked in red.
Reviewer 1
The title is not appropriate. This paper does not defines the optimal therapeutic dose as only one dose and 2 timings are assessed. Also there are, in this paper, no physiological read out that defines if the achieve concentrations are optimal. The fact the achieved concentrations are below 80 ng/mL is not sufficient to define these concentrations as optimal. The optimum 25(OH)VTD serum concentration is still controversial.
Response: We agree with the reviewer, and now we have changed the title to the following: „EFFECTS OF SUPPLEMENTATION IN VITAMIN D3 DEFICIENT OR INSUFFICIENT CHILDREN WITH ALLERGIC DISEASES “. Now concept of „optimal“ dose is removed.
It is misleading to compare timing of supplementation as different timings were applied to different populations. Conclusion on this point are doubtful.
Response: We agree with the reviewer, and in the revised version we deleted comparison of timing of supplementation in the Discussion section (first paragraph).
If data on corticosteroid are a key conclusive point of the study, it would be better to describe them more in details. It could be interesting to have the number of children in these groups and the 25-(OH)D levels before and after supplementation.
Response: We are grateful to the reviewer, because he/she has helped us to discover mistake in data interpretation. The coefficients B of multiple linear regression were negative for taking inhalational or nasal steroids, so we mistakenly thought that these prescriptions were associated with lower increase in Vid D levels. However, we overlooked that in the SPSS table children not taking steroids were designated with “2”, and those taking steroids with “1” (we thought wrongly that it was “0” and “1”, respectively). So, children not taking steroids had larger negative B coefficients, so their increase in 25(OH)D level was lower. Now we have corrected this mistake in the Results and Conclusions, including those of the Abstract, and also added the precise data about increase in 25(OH)D in children taking and not taking steroids:
“Children prescribed with inhalation corticosteroids (n = 36) had higher increase in the the 25-(OH)D levels than children without (n = 58) such prescription (19.3 ± 15.4 ng/ml vs. 13.4 ± 10.5 ng/ml, respectively). Children prescribed with nasal corticosteroids (n = 37) had higher increase in the the 25-(OH)D levels than children without (n = 57) such pre-scription (17.4 ± 14.7 ng/ml vs. 14.5 ± 11.5 ng/ml, respectively).”
Conclusions seem overstated. It would be better to write “children with allergic diseases can be treated with maximal recommended doses of vitamin D3 for a short period of time, especially…”
Response: We agree with the reviewer, and we have changed the conclusion accordingly: “In conclusion, vitamin D deficiency and insufficiency in children with allergic diseases can be treated with maximal recommended doses of vitamin D3 for a short period of time, especially children with allergic diseases should be treated with maximal recommended doses of vitamin D3, especially if they were prescribed with inhalation or intranasal corticosteroids.”
It would be easier to read if table 1 and 2 would be presented as a unique table with 25-(OH)D levels before and after the supplementation on the same line.
Response: It seems like this, but distribution of vitamin D deficiency/insufficiency was not the same before and after supplementation, so merging these two tables would make more complex and larger table that would not be easy to read and fit to the page. Therefore, we would prefer to remain with the two tables, as it is now.
Elecsys kits on cobas do not measure 25(OH)VTD3 but the total concentration of vitamin D (D2 + D3). Therefore, terminology should be adjusted and the abbreviation 25-(OH)D should be favored.
Response: We agree, so we changed the abbreviation accordingly.
The level of toxicity is sometimes defined as 100 ng/mL (line 64) and sometimes as 110 mcg/mL. For ease of reading, it is better to keep the same cut-off for the entire paper.
Response: We agree, so we set the level to 100 mg/ml in all parts of the manuscript.
The dose of Vitamin D is referred as 5,000IJ line 124 whereas it is referred as 4,000IU/day for the entire paper.
Response: Thank you for the remark. We now changed it to 4000 IU/day.
Line 227 did the author wanted to write “sample” or sample size”?
Response: Yes, we wanted to say “sample size”, so now it is corrected.
We hope that now the manuscript will be acceptable for publication.
Regards,
Prof Anđelka Stojković
Reviewer 2 Report
This is an important study that confirmed the efficacy and safety of 4000 IU/day vitamin D3 administration in children with allergic diseases. It is also novel to point out that there may be a difference in the effect of vitamin D supplementation depending on whether inhaled or nasal corticosteroids are used.
However, there are some points that have not been sufficiently considered, and there are some points where the contents described in the method differ from the contents described in the result, which need to be corrected.
Major points
- As for the timing of measurement of serum vitamin D levels after vitamin D administration, it is stated to be about 2 weeks, but please specify the actual time period between the end of vitamin D supplementation and blood collection with each participants, as it may affect the results.
- It seems that the confounders that were actually examined (Lines 166-170) are different from the confounders that have been included in the Method (Lines 90-97). In particular, what is described in the methods; (1) a season when serum 25-(OH)D3 level was measured (spring, autumn, or winter); (2) vitamin D supplementation that ended the latest 6 weeks before the onset of the last vitamin D3 supplementation regimen prescribed by a pediatricians are important as a confounding factors. Please check.
- In addition to fish and milk consumption, consider adding food allergies to these foods, as well as vitamin D-rich foods such as egg yolks and mushrooms, to the list of variables.
Minor points
- Include food allergy into entry criteria (Line83-84)
- Change”ad“ to “as” in line 102.
- Change ”mcg“ to “ng” in line 186.
Author Response
University of Kragujevac,
Faculty of Medical Sciences
To: Editors of Medicina
Dear Editors,
we are submitting our manuscript No medicina-1389149, entitled “OPTIMAL THERAPEUTIC DOSE IN VITAMIN D3 DEFICIENT OR INSUFFICIENT CHILDREN WITH ALLERGIC DISEASES” corrected according to the requests of the reviewers. The corrections in the manuscript are marked with “track changes” option. Below are corrections listed point-by-point according to requests of the reviewers, marked in red.
Reviewer 2
This is an important study that confirmed the efficacy and safety of 4000 IU/day vitamin D3 administration in children with allergic diseases. It is also novel to point out that there may be a difference in the effect of vitamin D supplementation depending on whether inhaled or nasal corticosteroids are used. However, there are some points that have not been sufficiently considered, and there are some points where the contents described in the method differ from the contents described in the result, which need to be corrected.
Major points
As for the timing of measurement of serum vitamin D levels after vitamin D administration, it is stated to be about 2 weeks, but please specify the actual time period between the end of vitamin D supplementation and blood collection with each participants, as it may affect the results.
Response: The time period was not “about”, but exactly 2 weeks. Writing “about” was consequence of inappropriate translation. Now we have changed this expression to exactly.
It seems that the confounders that were actually examined (Lines 166-170) are different from the confounders that have been included in the Method (Lines 90-97). In particular, what is described in the methods; (1) a season when serum 25-(OH)D3 level was measured (spring, autumn, or winter); (2) vitamin D supplementation that ended the latest 6 weeks before the onset of the last vitamin D3 supplementation regimen prescribed by a pediatricians are important as a confounding factors. Please check.
Response: It was writing error; the confounders mentioned in the Methods were taken into the model, but missed in the description of the Results. In the revised manuscript we have added them in the Results, too.
In addition to fish and milk consumption, consider adding food allergies to these foods, as well as vitamin D-rich foods such as egg yolks and mushrooms, to the list of variables.
Response: We have already included food allergy, as mentioned by skin prick test on nutritive allergens (n=20) – now we just added it to the Method, to make it more clear. Egg yolks and mushrooms are also added to the list.
Minor points
Include food allergy into entry criteria (Line83-84)
Response: We now added the food allergy to inclusion criteria.
Change”ad“ to “as” in line 102.
Reponse: Changed, as requested.
Change ”mcg“ to “ng” in line 186.
Response: Changed, as required.
We hope that now the manuscript will be acceptable for publication.
Regards,
Prof Anđelka Stojković
Round 2
Reviewer 1 Report
Major remarks have been addressed
Reviewer 2 Report
The authors have made appropriate corrections to the points raised.